

# Association of *HMGCR* rs17671591 and rs3761740 with lipidemia and statin response in Uyghurs and Han Chinese

Ziyang Liu[1,2,3,*], Yang Zhou[1,*], Menglong Jin[3], Shuai Liu[1], Sen Liu[1], Kai Yang[1], Huayin Li[1], Sifu Luo[1], Subinuer Jureti[1], Mengwei Wei[1] and Zhenyan Fu[1]

[1] The First Affiliated Hospital, Xinjiang Medical University, Urumqi, Xinjiang, China
[2] Xinjiang Medical University, Urumqi, Xinjiang, China
[3] State Key Laboratory of Pathogenesis, Prevention and Treatment of High Incidence Diseases in Central Asia, Xinjiang Medical University, Urumqi, Xinjiang, China
* These authors contributed equally to this work.

Corresponding author
Zhenyan Fu,
fuzhenyan316@126.com

## ABSTRACT

**Background:** Dyslipidemia plays a very important role in the occurrence and development of cardiovascular disease (CVD). Genetic factors, including single nucleotide polymorphisms (SNPs), are one of the main risks of dyslipidemia. 3-hydroxy-3-methylglutaryl-CoA reductase (HMGCR) is not only the rate-limiting enzyme step of endogenous cholesterol production, but also the therapeutic target of statins.

**Methods:** We investigated 405 Han Chinese and 373 Uyghur people who took statins for a period of time, recorded their blood lipid levels and baseline data before and after oral statin administration, and extracted DNA from each subject for SNP typing of *HMGCR* rs17671591 and rs3761740. The effects of *HMGCR* rs17671591 and rs3761740 on lipid levels and the effect of statins on lipid lowering in Han Chinese and Uyghur ethnic groups were studied.

**Results:** In this study, for rs17671591, the CC *vs.* TT+CT model was significantly correlated with the level of LDL-C before oral statin in the Uyghur population, but there were no correlations between rs17671591 and the level of blood lipid before oral statin in the Han population. The CC *vs.* TT+CT and CT *vs.* CC+TT models were significantly correlated with the level of LDL-C after oral statin in the Uyghur population. There was no significant correlation between rs3761740 with blood lipids before and after oral statin in the Han population. For rs3761740, before oral statin, the CC *vs.* AA+CA model was significantly correlated with the level of LDL-C, and the CA *vs.* CC+AA model was significantly correlated with the level of total cholesterol (TC), low density lipoprotein cholesterol (LDL-C), and non-high density lipoprotein cholesterol (HDL-C) in the Uyghur population. After oral statin, the CC *vs.* AA+CA and CA *vs.* CC+AA models were significantly correlated with the level of TC, LDL-C, and apolipoprotein (APOB), and the C *vs.* A model was significantly correlated with the level of TC, triglyceride (TG), LDL-C, and APOB in the Uyghur population. Particularly, the CT *vs.* CC+TT model of rs17671591 was significantly correlated with the changes of LDL-C after oral statin in the Uyghur population. In this study, we also explored the association of rs17671591 and rs3761740 with the rate of dyslipidemia as a reference.

**Conclusion:** We found that *HMGCR* rs3761740 was correlated with the levels of TC, LDL-C, and non-HDL-C before and after oral statin in Uyghurs, but not with blood lipid levels in the Han population. In the Uyghur population, *HMGCR* rs17671591 was associated with the level of LDL-C before and after oral statin, and also affected the changes of LDL-C after oral statin.

## INTRODUCTION

Cardiovascular disease (CVD) is the primary cause of human death, and constitutes significant health and economic burdens worldwide (*Tsao et al., 2023*; *Ralapanawa & Sivakanesan, 2021*; *Erbel et al., 2014*). Coronary heart disease kills more than 7.96 million people worldwide in 2006 and 9.48 million in 2016 (*GBD 2016 Causes of Death Collaborators, 2016*). CVD is a complex disease that involves multiple mechanisms and cell types, and is affected by many risk factors such as diabetes, hyperlipidemia, smoking, hypertension, lack of exercise, obesity, and heredity (*Visseren et al., 2021*). Dyslipidemia plays a very important role in the occurrence and development of CVD (*Arvanitis & Lowenstein, 2023*; *O'Malley et al., 2020*).

Dyslipidemia is a disease characterized by a series of lipid metabolic disorders, such as those involving abnormally elevated plasma levels of lipid and lipoprotein dysfunction (*Wang et al., 2018*; *Ahmad & Leake, 2019*). Dyslipidemia is mainly characterized by an increase in plasma total cholesterol (TC), low density lipoprotein cholesterol (LDL-C), triglyceride (TG), and a decrease in high density lipoprotein cholesterol (HDL-C) (*Smith, 2019*; *Xiao et al., 2019*). Dyslipidemia is a major risk factor for CVD according to data from 2009 to 2012, with more than 100 million American adults age 20 and older having total cholesterol levels of 200 mg/dL (5.17 mol /L) or higher, and nearly 31 million having total cholesterol levels of 240 mg/dL (6.20 mol /L) or higher (*Mozaffarian et al., 2016*). Genetic factors, including single nucleotide polymorphisms (SNPs), are a major risk factor for dyslipidemia (*Liu et al., 2017*; *Lu et al., 2017*).

Mammalian 3-hydroxy-3-methylglutaryl-CoA reductase (HMGCR) is an endoplasmic reticulum-localized glycoprotein consisting of a hydrophobic n-terminal domain that spans the cell membrane eight times and a large soluble C-terminal domain that projects into the cytoplasm (*Liscum et al., 1985*). HMGCR is a rate-limiting enzyme for cholesterol synthesis and is closely related to plasma cholesterol content (*Luo, Yang & Song, 2020*).

Statins are currently recognized as the primary treatment for lowering blood lipids (*Grundy et al., 2019*). HMGCR is both a rate-limiting enzymatic step for endogenous cholesterol production and a therapeutic target for statins, which reduce cholesterol production in the liver (*Istvan & Deisenhofer, 2001*). The reduction of cholesterol in the liver further leads to the upregulation of low-density lipoprotein receptors, which enhances the clearance of TC and LDL-C from the circulation, thereby reducing lipid levels (*Brown & Goldstein, 1997*). However, despite the effectiveness of statins, there is

significant variation in how individual patients respond to them (*Simon et al., 2006*). Pharmacogenetic studies have demonstrated the effect of genetic polymorphism on the wide variability of responses to statins observed in patients (*Reiner, 2014*; *Schmitz & Drobnik, 2003*). In addition, genome-wide association studies have identified several genetic variants associated with higher or lower responses to statin therapy, primarily in genes associated with cholesterol homeostasis (*Barber et al., 2010*). It has also been reported that *HMGCR* SNPS are associated with lipid levels after statin treatment (*Cuevas et al., 2016*).

HMGCR is the rate-limiting enzyme for cholesterol synthesis and the main target of statin therapy. In this study, *HMGCR* rs17671591 and rs3761740 were selected to evaluate their effects on blood lipid levels and the lipid-lowering effect of statin in Han Chinese and Uyghur populations.

## METHOD

### Ethical approval of the research protocol

This study was approved by the Ethics Committee of the First Affiliated Hospital of Xinjiang Medical University (number: 220525-06-2305A-Y1) and carried out in accordance with the principles of the Declaration of Helsinki. Each participant received written informed consent, including express permission to conduct DNA analysis and collect relevant clinical data.

### Subjects

This is a single-center prospective cohort study of the effects of *HMGCR* rs17671591 and rs3761740 on blood lipid levels before lipid-lowering therapy and statin responsiveness. In this study, patients who were hospitalized in the Heart Center of the First Affiliated Hospital of Xinjiang Medical University from 2008 to 2022 with long-term oral statins were selected, and patients' informed consent and signatures were required. Inclusion criteria: a) Uyghur or Han population aged 30–75; b) first diagnosed with coronary heart disease after admission; c) never took lipid-lowering medication before admission; d) started oral statin (atorvastatin 10 mg/day, rosuvastatin 5 mg/day) lipid-lowering treatment after admission; and e) follow up time ≥1 month. Exclusion criteria: a) patients with lost follow-up data; b) patients who adjusted their medication dosage or changed to different types of statins during the follow-up period; c) chronic renal insufficiency; d) fatty liver; e) cirrhosis; f) thyroid disease; or g) patients with genetic sequencing errors. The specific process is shown in Fig. 1. The hospital information platform was used to collect information of patients. Clinical data, blood lipid levels, and biochemical indexes of each patient were collected before oral statin and after oral statin administration for >1 month. The following information was collected: ethnicity, sex, age, blood glucose, alanine aminotransferase (ALT), TC, TG, HDL-C, LDL-C, apolipoprotein A1 (APOA1), apolipoprotein (APOB), lipoprotein a (Lpa), and non-HDL-C, where non-HDL-C = TC-HDL-C. All specimens collected were transported to the Xinjiang CHD VIP Laboratory on dry ice at predetermined intervals. Blood glucose, ALT, TC, TG, HDL-C, LDL-C, APOA1, APOB, and Lpa were all tested by the clinical laboratory of the First Affiliated Hospital of

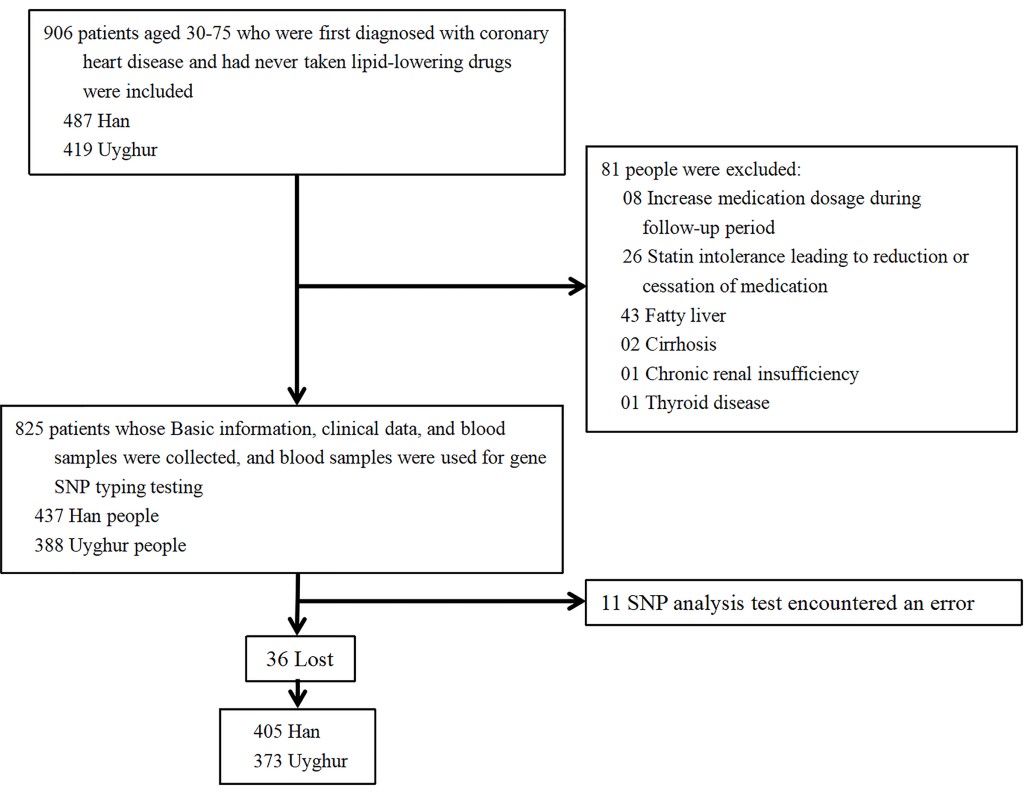

**Figure 1 Inclusion and exclusion process of research subjects.**

Xinjiang Medical University using a biochemical analyzer (Dimension AR/AVL Clinical Chemistry System, Newark, NJ, USA).

## Definition of related indicators

According to the 2023 Chinese Lipid Management Guidelines (*Guidelines for the Management of Blood lipids in China, 2023*), high TC was defined as TC ≥ 6.2 mmol/L, high TG was defined as TG ≥ 2.3 mmol/L, low HDL-C was defined as HDL-C < 1 mmol/L, high LDL-C was defined as LDL-C ≥ 4.1 mmol/L, low APOA1 was defined as APOA1 < 1.2 mmol/L, high APOB was defined as APOB > 1.1 mmol/L, high Lpa was defined as Lpa ≥ 300 mg/L, and high non-HDL-C was defined as non-HDL-C ≥ 4.9 mmol/L. Changes of lipids (TC, TG, HDL-C,LDL-C, APOA1, APOB, Lpa, non-HDL-C)=lipids before oral statin-lipid after oral statin)/(lipids before oral statin).

## Genotyping

DNA was extracted using the standard method, and then a custom-by-design 48-Plex SNPscanTM Kit (Cat#:G0104; Genesky Biotechnologies Inc., Shanghai, China) was used for SNP typing. According to the instructions of the kit, DNA underwent denaturation, ligation reaction, and PCR reaction. An ABI3730XL sequencer was used to separate and detect PCR products. The people who carried out genotyping did not know the baseline

data nor other subject indicators. To ensure the quality of genotyping, 4% of the DNA samples were taken again for repeated analysis.

## Statistical analysis

SPSS 25.0 was used for statistical analysis. The Hardy-Weinberg equilibrium test was performed by Chi-square test. Continuous variables were expressed as mean ± standard deviation, and differences between groups were assessed using an independent sample t-test or ANOVA. Comparison of lipid levels before and after statin treatment was analyzed using a paired sample t-test. The rate of categorical variables was expressed and Chi-square test was used to analyze the differences between groups. Multivariate analysis was further adjusted using linear regression models or logistic regression models, because age, sex, and liver function are important factors affecting blood lipids (*Grundy et al., 2019*), and adjustment variables included gender, age, and ALT. In addition, a two-tailed *P*-value of less than 0.05 was considered statistically significant.

# RESULTS

## Individual demographics

A total of 778 subjects were studied, including 405 Han and 373 Uyghur individuals. 75.34% of the Uyghur subjects were male, which was significantly higher than the 61.18% of Han subjects who were male. There were no significant differences in fasting blood glucose, TG, APOB, Lpa, and non-HDL-C between Han and Uyghur subjects before oral statin treatment. The age, TC, LDL-C, HDL-C, and APOA1 of Han subjects before oral statin treatment were significantly higher than those of Uyghur subjects before oral statin treatment. The ALT of Han subjects before oral statin treatment were significantly lower than those of Uyghur subjects before oral statin treatment (Table 1). In the Han population, oral statin significantly reduced the plasma concentrations of TC, TG, HDL-C, LDL-C, APOA1, APOB, and non-HDL-C, while oral statin did not significantly reduce the plasma concentrations of Lpa. In the Uyghur population, the plasma concentrations of TC, TG, LDL-C, APOA1, APOB, and non-HDL-C were significantly reduced after oral statin, while the plasma concentrations of HDL-C and Lpa were not significantly different before and after oral statin (Table 2).

## Effects of *HMGCR* rs17671591 (SNP1) on blood lipid levels before and after oral statin in Han and Uyghur populations

The distribution of SNP1 genotypes in both Han and Uyghur populations corresponded to the Hardy-Weinberg equilibrium. Before oral statin, there were no correlation between serum lipid levels and SNP1 in the Han population, but there were significant differences in LDL-C levels in the dominant model (CC and CT+TT) in the Uyghur population (Table S1). Before oral statin, the influence of the SNP1 dominant model (CC and TT +CT) on LDL-C plasma concentration was still statistically significant after multivariate adjustment in the Uyghur population (Table 3). After oral statin, there was a correlation between plasma APOA1 level and SNP1 in the Han population, and a correlation between plasma LDL-C level and SNP1 in the Uyghur population (Table S2). After

**Table 1 Clinical and metabolic characteristic of subjects.**

|  | Han ($n$ = 405) | Uyghur ($n$ = 373) | P |
|---|---|---|---|
| Sex (male) | 249 (61.18%) | 281 (75.34%) | <0.001 |
| Age (year old) | 60.290 ± 11.020 | 55.580 ± 8.938 | <0.001 |
| Fasting blood glucose (mmol/L) | 6.387 ± 2.607 | 6.426 ± 2.956 | 0.845 |
| TG (mmol/L) | 2.155 ± 1.334 | 2.295 ± 1.558 | 0.177 |
| TC (mmol/L) | 5.144 ± 0.946 | 4.911 ± 1.072 | 0.001 |
| HDL-C (mmol/L) | 1.118 ± 0.333 | 0.962 ± 0.255 | <0.001 |
| LDL-C (mmol/L) | 3.507 ± 0.752 | 3.342 ± 0.854 | 0.004 |
| APOA1 (mmol/L) | 1.241 ± 0.276 | 1.132 ± 0.220 | <0.001 |
| APOB (mmol/L) | 1.096 ± 0.253 | 1.061 ± 0.267 | 0.064 |
| Lpa (mg/L) | 237.836 ± 229.412 | 274.483 ± 289.213 | 0.051 |
| Non-HDL-C (mmol/L) | 4.026 ± 0.952 | 3.949 ± 1.058 | 0.284 |
| ALT (U/L) | 26.710 ± 20.578 | 33.310 ± 25.525 | <0.001 |

Note:
A t-test was conducted to generate the P values. Abbreviation: TC, total cholesterol; TG, triglycerides; HDL-C, high-density lipoprotein cholesterol; LDL-C, low-density lipoprotein cholesterol; APOA1, apolipoprotein A1; APOB, apolipoprotein B; Lpa, lipoprotein a; ALT, alanine aminotransferase.

**Table 2 Lipid profile of patients before and after atorvastatin therapy.**

| Ethnic group | Blood lipid | Before oral statin | After oral statin | P |
|---|---|---|---|---|
| Han ($n$ = 405) | TG (mmol/L) | 2.151 ± 1.342 | 1.895 ± 1.354 | <0.001 |
|  | TC (mmol/L) | 5.142 ± 0.951 | 3.873 ± 1.025 | <0.001 |
|  | HDL-C (mmol/L) | 1.122 ± 0.331 | 1.081 ± 0.292 | 0.017 |
|  | LDL-C (mmol/L) | 3.514 ± 0.753 | 2.352 ± 0.813 | <0.001 |
|  | APOA1 (mmol/L) | 1.241 ± 0.282 | 1.182 ± 0.253 | <0.001 |
|  | APOB (mmol/L) | 1.100 ± 0.252 | 0.830 ± 0.254 | <0.001 |
|  | Lpa (mg/L) | 237.703 ± 228.725 | 245.570 ± 248.344 | 0.437 |
|  | Non-HDL-C (mmol/L) | 4.037 ± 0.956 | 2.798 ± 1.044 | <0.001 |
| Uyghur ($n$ = 373) | TG (mmol/L) | 2.281 ± 1.522 | 1.952 ± 1.440 | <0.001 |
|  | TC (mmol/L) | 4.953 ± 1.062 | 3.992 ± 1.164 | <0.001 |
|  | HDL-C (mmol/L) | 0.992 ± 0.313 | 0.993 ± 0.282 | 0.761 |
|  | LDL-C (mmol/L) | 3.374 ± 0.840 | 2.524 ± 0.941 | <0.001 |
|  | APOA1 (mmol/L) | 1.151 ± 0.221 | 1.115 ± 0.276 | 0.004 |
|  | APOB (mmol/L) | 1.072 ± 0.267 | 0.874 ± 0.281 | <0.001 |
|  | Lpa (mg/L) | 265.750 ± 279.740 | 252.911 ± 270 | 0.304 |
|  | Non-HDL-C (mmol/L) | 3.963 ± 1.042 | 2.992 ± 1.153 | <0.001 |

Note:
A t-test was conducted to generate the P values. Abbreviation: TC, total cholesterol; TG, triglycerides; HDL-C, high-density lipoprotein cholesterol; LDL-C, low-density lipoprotein cholesterol; APOA1, apolipoprotein A1; APOB, apolipoprotein B; Lpa, lipoprotein a; ALT, alanine aminotransferase.

multivariate adjustment, the influence of the SNP1 allele model (C and T) after oral statin on APOA1 plasma concentration was still statistically significant in the Han population, and the influence of the SNP1 dominant model (CC and CT+TT) and additive model (CT and CC+TT) on LDL-C plasma concentration was still statistically significant in the Uyghur population (Table 3).

**Table 3 Association of SNP1 (rs17671591) and SNP2 (rs3761740) with blood lipid level before and after oral statin and after multivariate adjustment.**

| SNP | Statin | Ethnic | Blood lipids | Gene model | | β | P |
|---|---|---|---|---|---|---|---|
| SNP1 | Before oral statin | Uyghur | Blood lipids | CC (n = 142) | TT+CT (n = 231) | β | P |
| | | | LDL-C (mmol/L) | 3.220 ± 0.665 | 3.410 ± 0.945 | 0.190 | 0.038 |
| | After oral statin | Han | Blood lipids | CC (n = 179) | TT+CT (n = 226) | β | P |
| | | | APOA1 | 1.209 ± 0.253 | 1.156 ± 0.244 | −0.048 | 0.054 |
| | | | Blood lipids | TT (n = 36) | TT+CT (n = 369) | β | P |
| | | | APOA1 | 1.220 ± 0.219 | 1.240 ± 0.281 | 0.082 | 0.062 |
| | | | Blood lipids | C (n = 548) | T (n = 262) | β | P |
| | | | APOA1 | 1.194 ± 0.247 | 1.148 ± 0.252 | 0.041 | 0.024 |
| | | Uyghur | Blood lipids | CC (n = 142) | TT+CT (n = 231) | β | P |
| | | | LDL-C | 2.485 ± 0.815 | 2.704 ± 0.944 | 0.225 | 0.02 |
| | | | Blood lipids | TT (n = 56) | TT+CT (n = 317) | β | P |
| | | | LDL-C | 2.485 ± 0.815 | 2.704 ± 0.944 | 0.155 | 0.242 |
| | | | Blood lipids | CT (n = 175) | CC+TT (n = 198) | β | P |
| | | | LDL-C | 2.775 ± 0.923 | 2.485 ± 0.864 | −0.294 | 0.002 |
| SNP2 | Before oral statin | Uyghur | Blood lipids | CC (n = 326) | AA+CA (n = 47) | β | P |
| | | | LDL-C (mmol/L) | 3.290 ± 0.726 | 3.720 ± 1.415 | 0.431 | 0.001 |
| | | | Blood lipids | CA (n = 44) | AA+CC (n = 329) | β | P |
| | | | TC (mmol/L) | 5.420 ± 1.712 | 4.840 ± 0.937 | −0.579 | 0.001 |
| | | | LDL-C (mmol/L) | 3.790 ± 1.440 | 3.280 ± 0.724 | −0.499 | <0.001 |
| | | | Non-HDL-C (mmol/L) | 4.439 ± 1.659 | 3.883 ± 0.933 | −0.561 | 0.001 |
| | After oral statin | Uyghur | Blood lipids | CC (n = 326) | AA+CA (n = 47) | β | P |
| | | | TC (mmol/L) | 4.015 ± 1.108 | 4.532 ± 1.341 | 0.544 | 0.002 |
| | | | LDL-C (mmol/L) | 2.564 ± 0.847 | 3.023 ± 1.159 | 0.466 | 0.001 |
| | | | APOB (mmol/L) | 0.881 ± 0.262 | 1.007 ± 0.311 | 0.132 | 0.002 |
| | | | Blood lipids | CA (n = 44) | AA+CC (n = 329) | β | P |
| | | | TC (mmol/L) | 4.576 ± 1.37 | 4.013 ± 1.104 | −0.591 | 0.001 |
| | | | LDL-C (mmol/L) | 3.105 ± 1.154 | 2.557 ± 0.846 | −0.556 | <0.001 |
| | | | APOB (mmol/L) | 1.018 ± 0.312 | 0.88 ± 0.262 | −0.145 | 0.001 |
| | | | Blood lipids | C (n = 696) | A (n = 50) | β | P |
| | | | TG (mmol/L) | 1.932 ± 1.32 | 2.373 ± 1.64 | −0.486 | 0.012 |
| | | | TC (mmol/L) | 4.049 ± 1.132 | 4.493 ± 1.314 | −0.468 | 0.005 |
| | | | LDL-C (mmol/L) | 2.597 ± 0.877 | 2.952 ± 1.158 | −0.358 | 0.007 |
| | | | APOB (mmol/L) | 0.89 ± 0.267 | 0.997 ± 0.309 | −0.112 | 0.005 |

**Note:**
Multivariate analysis was further adjusted using linear regression models and adjustment variables including gender, age, and ALT. The dominant model takes the CC genotype as the reference, the recessive model uses the TT genotype as the reference, and the allele model uses the C allele as the reference. Abbreviation: LDL-C, low-density lipoprotein cholesterol; APOA1, apolipoprotein A1; ALT, alanine aminotransferase.

### Effects of *HMGCR* rs3761740 (SNP2) on blood lipid levels before and after oral statin in Han and Uyghur populations

The distribution of SNP2 genotypes in both Han and Uyghur populations corresponded to the Hardy-Weinberg equilibrium. No AA genotype was detected in the Han population, but only three individuals with the AA genotype were detected in the Uygur population.

Therefore, the dominant model (CC and CA) and allele model (C and A) were analyzed in the Han population, and the dominant model (CC and AA+CA), additive model (CA and CC+AA), and allele model (C and A) were analyzed in the Uyghur population. Before oral statin, there was no correlation between the blood lipids and SNP2 in the Han population, and there was a correlation between the LDL-C, TC, non-HDL-C, and SNP2 in the Uyghur population (Table S3). After multi-factor adjustment and before oral statin in the Uyghur populations, the influence of the SNP2 dominant model (CC and AA+CA) on LDL-C was still statistically significant, and the SNP2 additive model (CA and CC+AA) still had statistically significant effects on TC, LDL-C, and non-HDL-C (Table 3). After oral statin, there was no correlation between the blood lipids and SNP2 in the Han population, and there was a correlation between the TC, TG, LDL-C, APOB, and SNP2 in the Uygur population (Table S4). After multi-factor adjustment and oral statin in the Uyghur populations, the influence of the SNP2 dominant model (CC and AA+CA) and additive model (CA and CC+AA) on TC, LDL-C, and APOB was still statistically significant, and the influence of the SNP2 allele model (C and A) on TG, TC, LDL-C, and APOB was still statistically significant (Table 3).

## The effect of *HMGCR* rs17671591 (SNP1) on the rate of dyslipidemia before and after oral statin in Han and Uyghur populations

Before oral statin, there was a correlation between the rate of high TC and SNP1 in the Han population, and between the rates of high TC, low HDL-C, high LDL-C, and high non-HDL-C with SNP1 in the Uyghur population (Table S5). After multi-factor adjustment and before oral statin, the additive model (CT and CC+TT) of SNP1 still had statistical significance on the rate of high TC in the Han population. After multi-factor adjustment and before oral statin in the Uyghur population, the dominant model (CC and TT+CT) of SNP1 still had statistical significance in the rates of low HDL-C and high LDL-C in plasma, the recessive model (TT and CC+CT) of SNP1 still had statistical significance in the rates of high TC and high non-HDL-C in plasma, the additive model (CT and CC+TT) of SNP1 still had statistical significance in the rate of low HDL-C in plasma, and the allele model (C and T) of SNP1 still had statistical significance in the rates of high TC, low HDL-C, high LDL-C, high non-HDL-C in plasma (Table 4). After oral statin, there was a correlation between the rates of high APOA1 and SNP1 in the Han population, and a correlation between the rates of high LDL-C with SNP1 in the Uyghur population (Table S6). After multi-factor adjustment and oral statin, the dominant model (CC and TT+CT) and allele model (C and T) of SNP1 still had statistical significance on the rate of high APOA1 in the Han population. The dominant model and allele model of SNP1 still had statistical significance on the rate of high LDL-C in the Uyghur population (Table 4).

## The effect of *HMGCR* SNP2 (rs3761740) on the rate of dyslipidemia before and after oral statin in the Han and Uyghur populations

Before oral statin, there was no correlation between the rate of dyslipidemia and SNP2 in the Han population, and there was a correlation between the rate of high TC, high LDL-C, high non-HDL-C, and SNP2 in the Uyghur population (Table S7). After multi-factor

**Table 4 The effect of HMGCR SNP1 (rs17671591) and SNP2 (rs3761740) on the rate of dyslipidemia before and after oral statin and after multivariate adjustment.**

| SNP | Statin | Ethnic | Blood lipids | Gene model | | OR | P |
|---|---|---|---|---|---|---|---|
| SNP1 | Before oral statin | Han | Blood lipids | CT (n = 190) | CC+TT (n = 215) | OR | P |
| | | | High TC (%) | 17.277 | 8.796 | 0.415 | 0.005 |
| | | Uyghur | Blood lipids | CC (n = 142) | TT+CT (n = 231) | OR | P |
| | | | Low HDL-C (%) | 72.340 | 54.386 | 0.431 | <0.001 |
| | | | High LDL-C (%) | 7.746 | 14.719 | 2.070 | 0.047 |
| | | | Blood lipids | TT (n = 56) | CC+CT (n = 317) | OR | P |
| | | | High TC (%) | 19.643 | 8.833 | 0.394 | 0.018 |
| | | | High Non-HDL-C (%) | 25.000 | 12.618 | 0.401 | 0.011 |
| | | | Blood lipids | CT (n = 175) | CC+TT (n = 198) | OR | P |
| | | | Low HDL-C (%) | 54.286 | 67.677 | 1.878 | 0.004 |
| | | | Blood lipids | C (n = 459) | T (n = 287) | OR | P |
| | | | High TC (%) | 8.279 | 13.937 | 1.814 | 0.014 |
| | | | Low HDL-C (%) | 65.577 | 54.704 | 0.624 | 0.003 |
| | | | High LDL-C (%) | 10.022 | 15.331 | 1.637 | 0.030 |
| | | | High Non-HDL-C (%) | 12.418 | 17.770 | 1.559 | 0.035 |
| | After oral statin | Han | Blood lipids | CC (n = 179) | TT+CT (n = 226) | OR | P |
| | | | Low APOA1 (%) | 47.191 | 58.371 | 1.531 | 0.043 |
| | | | Blood lipids | C (n = 548) | T (n = 262) | OR | P |
| | | | Low APOA1 (%) | 50.368 | 59.843 | 0.705 | 0.028 |
| | | Uyghur | Blood lipids | CC (n = 142) | TT+CT (n = 231) | OR | P |
| | | | High LDL-C (%) | 2.113 | 8.696 | 4.514 | 0.017 |
| | | | Blood lipids | C (n = 459) | T (n = 287) | OR | P |
| | | | High LDL-C (%) | 4.585 | 8.741 | 0.483 | 0.018 |
| SNP2 | Before oral statin | Uyghur | Blood lipids | CC (n = 326) | AA+CA (n = 47) | OR | P |
| | | | High TC (%) | 7.975 | 27.660 | 4.403 | <0.001 |
| | | | High LDL-C (%) | 10.123 | 25.532 | 3.044 | 0.004 |
| | | | High Non-HDL-C (%) | 12.270 | 29.787 | 3.112 | 0.002 |
| | | | Blood lipids | CA (n = 44) | AA+CC (n = 329) | OR | P |
| | | | High TC (%) | 29.545 | 7.903 | 0.207 | <0.001 |
| | | | High LDL-C (%) | 27.273 | 10.030 | 0.299 | 0.002 |
| | | | High Non-HDL-C (%) | 31.818 | 12.158 | 0.291 | 0.001 |
| | | | Blood lipids | C (n = 696) | A (n = 50) | OR | P |
| | | | High TC (%) | 9.339 | 26.000 | 3.420 | <0.001 |
| | | | High LDL-C (%) | 11.207 | 24.000 | 2.506 | 0.009 |
| | | | High Non-HDL-C (%) | 13.506 | 28.000 | 2.557 | 0.005 |
| | After oral statin | Uyghur | Blood lipids | CC (n = 326) | AA+CA (n = 47) | OR | P |
| | | | High TC (%) | 3.988 | 13.043 | 4.190 | 0.007 |
| | | | Low HDL-C (%) | 61.963 | 45.652 | 0.528 | 0.046 |
| | | | High LDL-C (%) | 4.294 | 19.565 | 5.934 | <0.001 |
| | | | High APOB (%) | 18.650 | 39.130 | 2.965 | 0.001 |
| | | | High NonHDLC (%) | 5.828 | 17.391 | 4.058 | 0.003 |

(Continued)

| Table 4 (continued) | | | | | | | |
|---|---|---|---|---|---|---|---|
| SNP | Statin | Ethnic | Blood lipids | Gene model | | OR | P |
| | | | Blood lipids | C (n = 696) | A (n = 50) | OR | P |
| | | | High TC (%) | 4.604 | 12.245 | 0.305 | 0.013 |
| | | | Low HDL-C (%) | 60.863 | 46.939 | 1.732 | 0.067 |
| | | | High LDL-C (%) | 5.324 | 18.367 | 0.234 | <0.001 |
| | | | High APOB (%) | 20.000 | 38.776 | 0.379 | 0.002 |
| | | | High NonHDLC (%) | 6.619 | 16.327 | 0.317 | 0.007 |
| | | | Blood lipids | CA (n = 44) | CC+AA (n = 329) | OR | P |
| | | | High TC (%) | 13.953 | 3.951 | 0.219 | 0.005 |
| | | | Low HDL-C (%) | 44.186 | 62.006 | 1.997 | 0.037 |
| | | | High LDL-C (%) | 20.930 | 4.255 | 0.153 | <0.001 |
| | | | High APOB (%) | 39.535 | 18.790 | 0.332 | 0.002 |
| | | | High NonHDLC (%) | 18.605 | 5.775 | 0.221 | 0.001 |

**Note:**
Multivariate analysis was further adjusted using logistic regression models and adjustment variables including gender, age, and ALT. The additive model takes the CT genotype as the reference, the dominant model takes the CC genotype as the reference, the recessive model uses the TT genotype as the reference, and the allele model uses the C allele as the reference. Abbreviation: TC, total cholesterol; HDL-C, high-density lipoprotein cholesterol; LDL-C, low-density lipoprotein cholesterol; ALT, alanine aminotransferase.

adjustment in the Uyghur populations, the influence of the SNP2 dominant model (CC and AA+CA), additive model (CA and CC+AA), and allele (C and A) models on the rate of high TC, high LDL-C, and high non-HDL-C before oral statin was still statistically significant (Table 4). After oral statin, there was no correlation between the rate of dyslipidemia and SNP2 in the Han population, and there was a correlation between the rate of high TC, high LDL-C, high HDL-C, high APOB, high non-HDL-C, and SNP2 in the Uyghur population (Table S8). After multi-factor adjustment and oral statin in the Uyghur populations, the influence of the SNP2 dominant model (CC and AA+CA) and recessive model (CA and CC+AA) on the rate of high TC, low HDL-C, high LDL-C, high APOB, and high non-HDL-C was still statistically significant, and the influence of SNP2 allele model (C and A) on the rate of high TC, high LDL-C, high APOB, and high non-HDL-C was still statistically significant (Table 4).

### Effects of *HMGCR* rs17671591 (SNP1) and rs3761740 (SNP2) on the change of lipids after oral statin treatment in Han and Uyghur populations

There was no correlation between the change of lipids after oral statin and SNP2 in the Han and Uyghur populations (Table S9). After oral statin, there was a correlation between the change of APOA1 and SNP1 in the Han population, and a correlation between the changes of LDL-C and SNP1 in the Uyghur population (Table S10). After multi-factor adjustment and oral statin, the influence of the SNP1 alelle model (C and T) on the change of APOA1 was still statistically significant in the Han population, and the influence of the SNP1 additive model (CT and CC+TT) on the change of LDL-C was still statistically significant in the Uyghur population (Table 5).

**Table 5 Association of SNP1 (rs17671591) with changes of the blood lipids after multivariate adjustment.**

| Han | Change of lipids | C (548) | T (262) | β | P |
|-----|-----|-----|-----|-----|-----|
| | changes of APOA1 (%) | 1.591 ± 21.582 | 4.988 ± 20.683 | 3.56 | 0.028 |
| Uyghur | Change of lipids | CT (175) | CC + TT (198) | β | P |
| | changes of LDL-C (%) | 14.305 ± 35.39 | 21.059 ± 28.397 | 7.888 | 0.018 |

Note:
Multivariate analysis was further adjusted using linear regression models and adjustment variables including gender, age, and ALT. The additive model takes the CT genotype as the reference and the allele model uses the C allele as the reference. Abbreviation: LDL-C, low-density lipoprotein cholesterol; APOA1, apolipoprotein A1; ALT, alanine aminotransferase.

**Table 6 Association of SNP1 (rs17671591) with statin resistance rate in the Uyghur population.**

| | CT ($n$ = 175) | CC+TT ($n$ = 198) | Single factor analysis | | Multifactor adjustment | |
|-----|-----|-----|-----|-----|-----|-----|
| | | | OR | P | OR | P |
| Statin resistance | 64 (36.57%) | 54 (27.27%) | 0.6504 | 0.049 | 0.624 | 0.038 |
| Non-statin resistance | 111 (63.43%) | 144 (72.73%) | | | | |

Note:
Chi-square test was conducted to generate the $P$ values. Multivariate analysis was further adjusted using logistic regression models and adjustment variables including gender, age, and ALT. The additive model takes the CT genotype as the reference.

## Effect of *HMGCR* gene polymorphism on statin resistance

Samples with a decreased rate of plasma LDL-C concentration ≤10% after oral statin were set as the statin resistance group, and those with a decreased rate of LDL-C > 10% were set as the non-statin resistance group. It was found that the addition model of SNP1 had a statistically significant effect on the statin resistance rate in the Uyghur population (Table 6). After adjusting for multiple factors, the additive model of SNP1 in the Uyghur population still had a statistically significant effect on the statin resistance rate (Table 6).

## DISCUSSION

In this project, we mainly studied the effects of rs17671591 and rs3761740 of *HMGCR* on plasma lipid levels before and after oral statin in Han Chinese and Uyghur populations, as well as the effects on lipid lowering response after oral statin in two populations.

Mammalian HMGCR is an ER-localized glycoprotein consisting of a hydrophobic N-terminal domain that spans the membrane eight times, large hydrophilic C-terminal domain located on the cytoplasmic side, transmembrane domains 2–6 that function as a sterol-sensing domain (SSD) that sensitizes HMGCR to sterol levels in the endoplasmic reticulum, and a cytoplasmic C-terminal domain that is responsible for converting HMG-CoA to valerate using two NADPH molecules as reducing agents (*Liscum et al., 1985*). HMGCR plays one of the most important roles in cholesterol biosynthesis and is one of the two rate-limiting enzymes in cholesterol biosynthesis. It is also regulated by many factors and is crucial for maintaining lipid homeostasis in the body (*Luo, Yang & Song, 2020*). HMGCR is the main target of statins for lipid reduction (*Istvan & Deisenhofer, 2001*).

Some studies have proved that effective lipid changes can be seen in the 4th week of atorvastatin dose range of 10–80 mg (*Jones et al., 2005*), and this result has also been proved in other studies (*Szapary et al., 2004*) where the lipid level remained basically stable after 4 weeks (*Cannon et al., 2004*; *Niemi et al., 2004*), and so this study selected patients who took oral statin for more than 1 month as the study object. In the Clinical Pharmacogenetics Implementation Consortium (CPIC) guidelines, the dosing recommendations were provided according to SLCO1B1, ABCG2, and CYP2C9 genotypes. A dose of atorvastatin at 10 mg once daily and a dose of rosuvastatin at 5 mg once daily is associated with the lowest risk for statin-associated musculoskeletal symptoms (*Cooper-DeHoff et al., 2022*), so we chose patients who took 10 mg of atorvastatin daily or 5 mg of rosuvastatin daily as subjects. In this study, we confirmed that oral statin can effectively reduce TC, TG, LDL-C, and APOB, which is consistent with the results of *Jones et al. (2005)*. However, the effect of statins on HDL-C is controversial. *Jones et al.*'s *(2005)* study proved that statins can increase HDL-C, while *Szapary et al.*'s *(2004)* study proved that statin treatment does not affect the plasma concentration of HDL-C. This difference may be related to the dose of statin or the different ethnic genetic background of the study subjects. In this study, it was proved that the plasma HDL-C concentration of the Han population was significantly reduced after statin treatment, while the plasma HDL-C concentration of the Uyghur population was not significantly changed after statin treatment. Our study also proved that statins can effectively reduce plasma levels of TG, TC, LDL-C, APOA1, APOB, and non-HDL-C in both Han and Uyghur subjects.

The rs17671591 polymorphism is an intergenic variation located near the *HMGCR* gene promoter region (chr5:74,615,021); however, it is not directly located in the gene expression regulatory element or the region affecting the enzyme structure or conformation. The dominant model of rs17671591 polymorphism is associated with HDL-C plasma levels and affects the change of LDL-C and HDL-C plasma concentrations after statin treatment (*Cuevas et al., 2016*). The Treating to New Targets (TNT) study also showed an association between *HMGCR* rs17671591 and statin response (*Thompson et al., 2009*), but a genome-wide association (GWAS) study also proved that rs17671591 polymorphism was not associated with low density lipoprotein level after taking atorvastatin (*Deshmukh et al., 2012*). In this study, we found that the *HMGCR* rs17671591 genotype was associated with the rate of high TC before oral statin, APOA1 and the rate of high APOA1 after oral statin, and the change of APOA1 after oral statin in Han subjects. In the Uyghur population, we found that the *HMGCR* rs17671591 genotype was associated with LDL-C, low HDL-C, high LDL-C, high TC, and high non-HDL-C before oral statin; LDL-C, and high LDL-C after oral statin; and the change of LDL-C after oral statin. The difference of rs17671591 effects between the two ethnic groups may be due to their different genetic backgrounds.

The rs3761740 polymorphism is a variant located in the upstream region of the *HMGCR* gene chr5:75336308 (GRCh38.p14), but it not directly located in gene expression regulatory elements or regions affecting enzyme structure or conformation. Previous studies have not found a correlation between rs3761740with LDL-C and statins response

to LDL-C effect (*Angelini et al., 2017*). However, the results of this study are not completely consistent with this. In the Han population, there was no significant correlation between rs3761740 with blood lipids and lipid-lowering effect of statins. In the Uyghur population, we found that the *HMGCR* rs3761740 genotype was associated with TC, LDL-C, non-HDL-C, the rate of high TC, the rate of high LDL-C, the rate of high non-HDL-C before oral statin; and TC, TG, LDL-C, APOB, highTC, high LDL-C, low HDL-C, high APOB, and high non-HDL-C after oral statin.

The lipid-lowering effect of statins is different for each person. Many genetic mutations are related to the effectiveness of statins. These genes can be divided into two groups. First are those that are directly related to lipoprotein metabolism and affect either LDL production or its catabolism. The second group are drug metabolism–related genes that affect statin pharmacokinetics. Some studies have proved that the genetic variation of *HMGCR* is related to the resistance of statins, but it is not certain if it is due to the ability of statins to inhibit the enzyme or from the greater compensatory upregulation of the *HMGCR* gene (*Sun et al., 2023*). High-dose statin therapy is associated with lower blood lipids (*Li et al., 2021*), but the risk of intolerance with high-dose statins is also increased (*Cooper-DeHoff et al., 2022*). It is very important to predict the lipid-lowering effect of statins and choose low-dose statins that can effectively reduce blood lipids according to the predicted results. In our study, we found that the *HMGCR* rs17671591 additive model (CT and CC+TT) was associated with the change of LDL-C after oral statin in the Uyghur population. This SNP can be used to predict the efficacy of statins in the Chinese Uyghur population and provide strategies for precision therapy.

Different genetic models are supported by their assumed inheritance pattern, although no optimal models have been established (*Horita & Kaneko, 2015*). The levels of significance obtained from different models may be similar, but not exactly the same (*Horita & Kaneko, 2015*; *Clarke et al., 2011*). Although the analysis of multiple models may lead to concerns about multiple comparisons, the unified direction and association obtained may further support the association between SNP and disease status (*Horita & Kaneko, 2015*; *Wu et al., 2015*). In this study, different gene models were established to verify the relationship between each gene model, blood lipids, and statin response, and the consistent influence on the direction and association of each model can better support the conclusions.

The differences between this study and previous studies and between Han and Uyghur populations in this study are not surprising, as lipid concentrations are a complex feature of environmental and genetic factors and their interactions. Because genetic studies rely primarily on genetic diversity, other unexplained interactions are often excluded, including traits such as age, sex, ethnic origin, sample size, heterogeneity among patients, study design, use of different endpoints, evaluation time, drug dosage, strict inclusion/exclusion criteria, and underlying disease. As a result, it becomes very difficult to draw the same conclusions after comparing different studies, so the results tend to apply only to specific populations.

This study also has some limitations. First of all, the multivariate adjustment factors included in this study only included age, sex, and liver function as possible risk factors, but

did not include other potential risk factors affecting blood lipids including diet, exercise, smoking, drinking, and BMI. Second, the follow-up time of this study was ≥1 month and was not fixed, but we ensured that each subject did not change the lipid-lowering drugs and dosage during the study period. Some studies have proved that blood lipids tend to be stable after oral statin for 4 weeks (*Cannon et al., 2004*; *Niemi et al., 2004*), so the unfixed time had little effect on blood lipids after taking drugs.

## CONCLUSION

In this article, we mainly studied the effects of *HMGCR* rs17671591 and rs3761740 on plasma lipid levels before and after oral statin in Han Chinese and Uyghur populations, as well as the effects on lipid lowering response after oral statin in two populations. We found that the SNP of *HMGCR* was associated with the lipid-lowering effect of statins and provides the possibility to predict the reactivity of statins in different individuals.

## ACKNOWLEDGEMENTS

We thank all of the patients for participating in the study.

### Funding

This work was supported by National key Research and Development program of China [grant number: 2021YFC2500605]; The National Natural Science Foundation of China [grant numbers: 81970380]; State Key Laboratory of Pathogenesis, Prevention and Treatment of High Incident Disease in Central Asia [grant numbers: xyd2021C002]; State Key Laboratory of Pathogenesis, Prevention and Treatment of High Incidence Diseases in Central Asia Fund [SKL-HIDCA-2022-XXG3]. State Key Laboratory of Pathogenesis, Prevention and Treatment of High Incidence Diseases in Central Asia Fund [SKL-HIDCA-2023-36]. The funders had no role in study design, data collection and analysis, decision to publish, or preparation of the manuscript.

### Grant Disclosures

The following grant information was disclosed by the authors:
National Key Research and Development Program of China: 2021YFC2500605.
National Natural Science Foundation of China: 81970380.
State Key Laboratory of Pathogenesis.
Prevention and Treatment of High Incident Disease in Central Asia: xyd2021C002, SKL-HIDCA-2022-XXG3, SKL-HIDCA-2023-36.

### Competing Interests

The authors declare that there are no competing interests associated with the manuscript.

### Author Contributions

- Ziyang Liu conceived and designed the experiments, analyzed the data, authored or reviewed drafts of the article, and approved the final draft.

- Yang Zhou conceived and designed the experiments, analyzed the data, authored or reviewed drafts of the article, and approved the final draft.
- Menglong Jin performed the experiments, prepared figures and/or tables, and approved the final draft.
- Shuai Liu analyzed the data, prepared figures and/or tables, and approved the final draft.
- Sen Liu analyzed the data, prepared figures and/or tables, and approved the final draft.
- Kai Yang performed the experiments, prepared figures and/or tables, and approved the final draft.
- Huayin Li performed the experiments, prepared figures and/or tables, and approved the final draft.
- Sifu Luo performed the experiments, prepared figures and/or tables, and approved the final draft.
- Subinuer Jureti performed the experiments, prepared figures and/or tables, and approved the final draft.
- Mengwei Wei performed the experiments, prepared figures and/or tables, and approved the final draft.
- Zhenyan Fu conceived and designed the experiments, authored or reviewed drafts of the article, and approved the final draft.

## Human Ethics

The following information was supplied relating to ethical approvals (*i.e.*, approving body and any reference numbers):

The Ethics Committee of Xinjiang Medical University (number: 220525-06-2305A-Y1).

## Data Availability

The raw measurements are available in the Supplemental Files.

## Supplemental Information

Supplemental information for this article can be found online at http://dx.doi.org/10.7717/peerj.18144#supplemental-information.

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
