# Peer review of "Association of HMGCR rs17671591 and rs3761740 with lipidemia and statin response in Uyghurs and Han Chinese"

_PeerJ, doi:10.7717/peerj.18144_

## Round 0.1 · original submission · Major Revisions

· Academic Editor

Major Revisions

One reviewer suggested major revisions and clarifications to the manuscript.
Please address all comments from the reviewer and the following:

Table 1: In Table footnotes explain what type of a test was conducted to generate the P values

Table 2:
--In Table and in other Tables change "Ethic" with "Ethnic Group".
--In Table footnotes explain what type of a test was conducted to generate the P values

Table 3:
--Better to present in landscape (wide) format as the columns are very narrow for proper presentation
-- Present Hardy-Weinberg results in table footnotes and give more space to other columns on the table

Table 4:
--Better to present in landscape (wide) format as the columns are very narrow for proper presentation
-- Present Hardy-Weinberg results in table footnotes and give more space to other columns on the table

Table 5: Needs to be prepared again. Not clear what is compared to what and in which group
What factors are adjusted for? Explain in table footnotes.

Table 6:
--Better to present in landscape (wide) format as the columns are very narrow for proper presentation
-- Present Hardy-Weinberg results in table footnotes and give more space to other columns on the table

Table 7:
--Better to present in landscape (wide) format as the columns are very narrow for proper presentation
-- Present Hardy-Weinberg results in table footnotes and give more space to other columns on the table

Table 8: Needs to be prepared again. Not clear what is compared to what and in which group
What factors are adjusted for? Explain in table footnotes.

Table 11:
--What factors are adjusted for? Explain in table footnotes.
--Why do you write (%) sign in "rate of changes of APOA1(%)" and in "rate of changes of LDL-C(%)" The table does not present percentages, only metabolite concentrations are presented.

Table 12:
--What factors are adjusted for? Explain in table footnotes.

-- Throughout the manuscript, HMGCR should be italicized when used as a gene name. Such as HMGCR rs17671591

-- Why did you not use the other genotype data generated by the genotyping platform? The way the manuscript is written gives a message as if you "cherry picked" the meaningful SNP and only reported interesting results.

**Language Note:** The review process has identified that the English language must be improved. PeerJ can provide language editing services - please contact us at [email protected] for pricing (be sure to provide your manuscript number and title). Alternatively, you should make your own arrangements to improve the language quality and provide details in your response letter. – PeerJ Staff

Reviewer 1 ·

Basic reporting

Probably the findings and the study has some interesting points; however, it is really difficult to follow the text. The grammar and style of the manuscript must be improved, so it can be comprehensive. It also presents many information, unstructured, that make it confused. I hope the following comments could help the authors:
1. The title is too long. The word SNPs could be delated, probably the rs could not be included. I think that the phrase "Uyghur and Han of Chinese" is correct. It needs also a style correction.
2. The HMGCR gene must be italicized throughout the text.
3. The population name Uyghur is wrong written in the Abstract and Table 10.
4. The use of capital letters after points and comma is wrong employed. There are several capital letters after comma, and smal letters after points. A space is need after a point. In addition, there are different letter style in the text.
5. The term hyperTC should be defined the first time it is mentioned.
6. The Results section requires to be summarized and to improve the presentation of the findings. Some tables could be added as a supplementary material. The information contained in Tables should not be duplicated in the text.
7. There are several typos in the text. For instance: line 62 is series or serie?, line 72 A or a?, line 85 primarily in?, line 86 HMGCR-associated SNPs?, line 91 the word respectively is wrong employed. Line 128 adopted?
8. The sentence in lines 62-63 is not clear.
9. Subjects section should be better written. Verify for punctuation. The sentence in lines 102-103 includes the word hospitalized twice. Consider to present the inclusion criteria employing a), b), c), etc.
10. Line 111: Do authors refer to renal function?
11. Sentence in lines 116-118 is difficult to follow and the information is repeated.
12. The ideas in lines 160-162 could be reduced in one sentence.
13. The authors found that there were no differences in fasting glucose, but this was considere as a covariate for the adjustment. Please clarify.
14. All the tables require to be improved and all of them should include the information to make them self-explanatories. A footnote including the meaning of the abbreviations and the covariates included in the adjustment of the model should be added. The number between brackets considering the number of subjects in each population should include n= (i.e., n=405).
15. Table 1, please correct fastimg.
16. Line 176-177, the sentence needs to be clarified. The genotypes would not be modified with the statin treatment. You can express that those are the allele and genotype frequencies of the studied populations.
17. The tables should be ordered according they are mentioned in the text. For instance, the table 5 is cited before the table 4, so they should be reordered.
18. The terms hyperlipidemia and dyslipdemia are employed, please be consistent with the one correct term.
19. Lines 239 and 241: there are some symbols
20. Line 254: What do you mean with invisible model?
21. The sentence in lines 256-258 needs to be revised.
22. In table 8 could you explain why there are two OR values for each analyzed variable?

Experimental design

The study lacks of rigor in different aspects. Authors should considered the following comments:

1. There is a pharmacogenomic guideline recommendation for statin at CPIC, in which the dosing recommendations are provided according to SLCO1B1, ABCG2, and CYP2C9 genotypes ( doi:10.1002/cpt.2557). This guideline must be cited and authors should better justify the study of the two HMGCR variants included and the relevance of the studied Chinese populations.
2. According to the study design, an analysis for repeated measures is required.
3. How do the authors select the variables for multivariate adjustment? The selection of the covariates should be explained.
4. The authors stated that "the SNP genotyping work was performed using a custom-by-design 48-Plex SNPscanTM", but they only present the results of two variants. What did happen with the remaining variants?
5. The genetic models studies need to be revised. I suggest that authors justify their presentation of the genetic models according to the literature. Some available references are http://dx.doi.org/10.1016/j.mgene.2015.04.003 and doi: 10.1111/jcmm.12751.

Validity of the findings

The findings presentation should be improved. Authors found the association of a HMGCR variant with lipid levels and with the statin effect in the decrease of lipid levels. However, the presentation of the findings difficults the comprehension and decrease the value of the work.

Several results are repeated in the Discussion section (Lines 296-300 and 311-323). Authors should modify the Discussion to consider the relevance of the findings and the comparison with previous results, how the Chinese population could benefit of the finding, the perspectives and limitations of the study, the implication of other variants in this or other genes that could affect the statin response, the relation of non-genetic factors in the treatment response, etc.

Additional comments

No comment.

---

## Round 0.2 · Minor Revisions

· Academic Editor

Minor Revisions

Authors should address the points raised by the reviewer.

Reviewer 1 ·

Basic reporting

The manuscript still requires a grammar style correction. There are sentences that are not clear, and also the punctuation is not used adequately. In addition, the results in the Abstract should be summarized and improve its presentation to a better presentation of the main findings. The Results Section is also too long, some Tables could be merged in one Table, and different Results can be summarized to make the manuscript more clear and with a better comprehension of the Results.

Experimental design

The manuuscript was improved in this regard. Only verify that in some tables de OR is presented and beta value in others.

Validity of the findings

No comment.

Additional comments

No comment.

---

## Round 0.3 · accepted · Accept

· Academic Editor

Accept

Congratulations on the acceptance of your manuscript.

Please follow directions from the journal office for timely publication of your manuscript.